# Slow light nanocoatings for ultrashort pulse compression

M. Ossiander [1✉], Y.-W. Huang [1,2], W. T. Chen [1], Z. Wang [3], X. Yin[1], Y. A. Ibrahim[1,4], M. Schultze [3] & F. Capasso [1✉]

Transparent materials do not absorb light but have profound influence on the phase evolution of transmitted radiation. One consequence is chromatic dispersion, i.e., light of different frequencies travels at different velocities, causing ultrashort laser pulses to elongate in time while propagating. Here we experimentally demonstrate ultrathin nanostructured coatings that resolve this challenge: we tailor the dispersion of silicon nanopillar arrays such that they temporally reshape pulses upon transmission using slow light effects and act as ultrashort laser pulse compressors. The coatings induce anomalous group delay dispersion in the visible to near-infrared spectral region around 800 nm wavelength over an 80 nm bandwidth. We characterize the arrays' performance in the spectral domain via white light interferometry and directly demonstrate the temporal compression of femtosecond laser pulses. Applying these coatings to conventional optics renders them ultrashort pulse compatible and suitable for a wide range of applications.

[1] John A. Paulson School of Engineering and Applied Sciences, Harvard University, 29 Oxford St, Cambridge, MA 02138, USA. [2] Department of Photonics, National Yang Ming Chiao Tung University, Hsinchu 30010, Taiwan. [3] Institute of Experimental Physics, Graz University of Technology, Petersgasse 16, 8010 Graz, Austria. [4] University of Waterloo, Waterloo, ON N2L 3G1, Canada. ✉email: mossiander@g.harvard.edu; capasso@seas.harvard.edu

Femtosecond light pulses are the basis for the highest achievable time resolutions and electrical field intensities today and have become central tools in microscopy[1], medicine[2], technology[3], and physical chemistry[4]. A key challenge in their application remains dispersion control: because all transparent materials are normally dispersive in the ultraviolet, visible, and near-infrared regions below a wavelength of 1.3 μm, the realization of compressed laser pulses currently requires complex angular-dispersive[5–7], reflective[8–10], or photonic-crystal-fiber-based compression setups[11], which all add significant complexity, path length, and beam deviations to the optical setup. Recently, dielectric metasurfaces, in addition to their use as lenses and for spatially shaping the phase and polarization of light on the nanoscale[12–15], were employed in an angular-dispersive Fourier-transform setup[16], providing fine-grained control of the time-domain properties of ultrashort laser pulses.

In this work, we demonstrate ultrathin nanocoatings that induce anomalous group delay dispersion directly upon transmission. The approach is illustrated in Fig. 1a: the nanocoatings can straightforwardly be applied to conventional optics and compensate for their group delay dispersion or be implemented in existing laser setups to compress ultrashort laser pulses. As such, the coatings can simplify the use and expand the applicability of femtosecond laser pulses. Therefore, they can act as the basis for an array of anti- or non-dispersive optics. Compared to theoretically proposed approaches based on plasmonics[17] or flat nanodisk Huyghens metasurfaces[18], our high-aspect-ratio nanopillar coating combines high transmission, anomalous dispersion, low high-order dispersion, and broadband operation.

The influence of transmissive optics on the time-domain profile of ultrashort laser pulses can be quantified by the frequency-dependent group delay GD $= \frac{d\varphi}{d\omega}$, which is calculated as the derivative of the angular-frequency-dependent spectral phase $\varphi(\omega)$ imprinted by the optics. Visible and near-infrared ultrashort laser pulses transmitted through transparent optics elongate because the pulses' high-frequency (blue) components are delayed more than their low-frequency (red) components, i.e., $\mathrm{GD}(\omega_\mathrm{red}) < \mathrm{GD}(\omega_\mathrm{blue})$. Far from resonances, most materials' group delay profiles are approximately linear, thus, they are approximated by their positive slope, the group delay dispersion GDD $= \frac{d\mathrm{GD}}{d\omega} = \frac{d^2\varphi}{d\omega^2} > 0$. To compensate for the temporal broadening of ultrashort pulses upon transmission through optical elements, our goal is to create a coating with the opposite effect, i.e., $\mathrm{GD}(\omega_\mathrm{red}) > \mathrm{GD}(\omega_\mathrm{blue})$ or GDD < 0.

## Results and discussion

**Working principle.** Our approach for creating a transmissive compressor coating uses uniform circular amorphous silicon nanopillars arranged in a periodic square array (see Fig. 1b). Its working principle can be approached from a scattering perspective[18–20] or from the perspective of an array of waveguides[21–24]. Here we follow the second approach.

We examine the dispersion of the eigenmodes of a two-dimensional compressor cross-section (i.e., the out-of-plane dispersion of the two-dimensional photonic crystal) in Fig. 1c. At near-infrared wavelengths, incoming light couples predominantly to two modes (see Fig. 1d) - with electric field profiles similar to the HE11 and HE12 hybrid modes in dielectric waveguides—due to their matching field symmetries (see Fig. 1e and ref. [21]). Light in the HE11-like mode is mainly confined in the silicon nanopillars whereas it also travels in free space in the HE12-like mode (see Fig. 1e). Because the compressor is large compared to its periodicity, light leaking out of a single nanopillar is not lost. Thus, the propagation constant of the HE12-like mode is real above the vacuum light line (see Fig. 1c).

The anomalous GDD of our device is caused by the dispersion of the HE12-like mode close to $k_z = 0$ where $k_z$ is the wavevector component in propagation direction: because light must travel at the same speed in both the backward and forward direction, reciprocity requires that the group velocity $v_g = \frac{d\omega}{dk_z} \to 0^+$ of light propagating in the HE12-like mode vanishes at $k_z \to 0^+$. In the absence of losses, this is true for all modes except for the fundamental mode or the special case of degenerate modes[25]. Consequently, close to the cutoff at $k_z = 0$, the mode enters a region of slow light[26], which is visible in Fig. 1c as the vanishing slope of the HE12-like mode dispersion. The bending of the dispersion from the vacuum light line at high frequencies into this slow-light region (see arrows in Fig. 1c) creates anomalous group-velocity dispersion GVD $= \frac{d^2 k_z}{d\omega^2} = \frac{dv_g^{-1}}{d\omega}$ at frequencies above $k_z = 0$. If light would couple exclusively to the HE12-like mode, the compressor group delay would diverge close to the $k_z = 0$ cutoff, preventing broadband operation. However, at the cutoff, light coupling rapidly shifts to the HE11-like mode (see Fig. 1d). The mixing of both modes generates a broadband region of constant anomalous GDD.

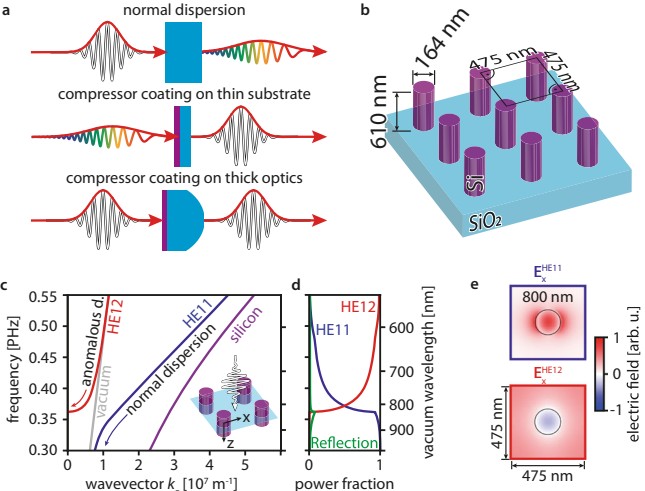

**Fig. 1 Nanocoating compressor working principle and design. a** Chromatic dispersion effect on ultrashort laser pulses in transparent materials and possible coating applications. Short pulses are marked in white, pulses stretched by material dispersion in colors; normally dispersive materials (light blue) elongate short pulses. The coating (purple) can either be applied to a thin substrate to shorten elongated pulses or to a thick optics to compensate for its group delay dispersion. **b** Design optimized for operation at 800 nm wavelength. **c** Dispersion of the predominant modes calculated for a two-dimensional cross-section (illustrated in the inset) perpendicular to the light propagation direction. The modes are labeled with the names of hybrid fiber modes with the same symmetry (HE11-like: blue line, HE12-like: red line). The vacuum (gray line) and silicon dispersions (purple line) are also plotted. One should note that the HE12-like mode's propagation constant is real above the vacuum line because light leaking out of a single nanopillar is not lost from the array. The mode's wavevector along the propagation direction $k_z$ vanishes ($k_z = 0$) at a frequency of 0.36 PHz (825 nm). As this cutoff is approached, the group velocity decreases (slow light), creating a region with anomalous group velocity dispersion (see red arrow marked anomalous d.) **d** Power coupling fraction from free space to the HE11-like and HE12-like modes. The transition from light propagating in the HE12-like to the HE11-like mode at 0.36 PHz (825 nm) increases the working bandwidth. Power reflected by the compressor is plotted in green. **e** Transverse electric field distributions of the modes at 800 nm wavelength. Whereas the HE11-like mode has no inflection points, the HE12-like mode has one inflection point.

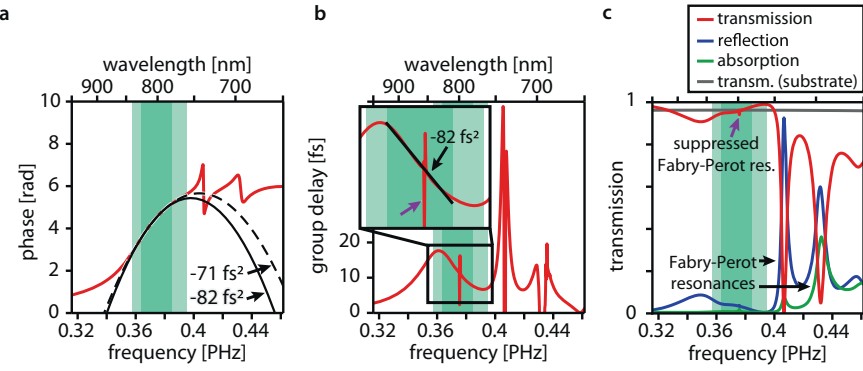

**Fig. 2 Finite-difference time-domain (FDTD) modeling results for the compressor coating. a** Transmission phase profile (red line) predicted by FDTD simulations for the compressor design displayed in Fig. 1b. The dashed black line is a parabolic fit to the phase profile within the full working range (light green area) and the solid black line is a parabolic fit to the phase profile within the linear working range (dark green area). The fits are labeled with the curvatures of the parabolae, which is the group delay dispersion. The downward curvature of the parabolae indicates anomalous dispersion. **b** Simulated transmission group delay profile (red line), i.e., the derivative of the transmission phase profile. The inset shows the magnified working ranges. The black line represents a linear fit to the linear working range (dark green area) and is labeled by its group delay dispersion. The downward slope indicates anomalous dispersion. In the full working range (light green region), the group delay deviates by <2 fs from a linear profile. A narrowband (<1 nm) residual signature of the suppressed Fabry-Perot resonance in the working range is marked with a purple arrow in **b** and **c** (see text and Supplementary Figs. 1 and 2). **c** Simulated transmission (red line), reflection (blue line), and linear absorption characteristics (green line) of a compressor-coated silicon dioxide substrate surface. Two-photon absorption[41] in the compressor coating remains below 1% for incident intensities beyond 50 GW cm$^{-2}$. The transmission of the uncoated surface is plotted as a reference (gray line). The working ranges (green areas) are marked as a guide to the eye.

The lower limit GDD$_{min}$ for the achievable constant anomalous GDD in a transmissive slow-light coating can be estimated similar to the maximum achievable delay in slow-light waveguides or scatterers[26,27]. For a given working range $\omega \in ]\omega_0 - \frac{\Delta\omega}{2}, \omega_0 + \frac{\Delta\omega}{2}] = ]\omega^-, \omega^+]$ defined by the cutoff $k_z(\omega^-) = 0$ and bandwidth $\Delta\omega$ (or the working range's central frequency $\omega_0$ and its high-frequency limit $\omega^+$), we find (see methods) that the achievable constant anomalous GDD is set by the thickness of the coating $L$ and the effective refractive index of the HE12 mode on the high-frequency side of the working range $n^+ = n(\omega^+) = \frac{k_z * c}{\omega}|_{\omega=\omega^+}$ (vacuum speed of light $c$):

$$\text{GDD}_{min} = L * \text{GVD}_{min} = \frac{L}{c\Delta\omega}\left(2 - n^+ - 2\frac{n^+ * \omega_0}{\Delta\omega}\right) \quad (1)$$

As example, using $n^+ \approx 1$ as suggested by Fig. 1c and choosing a working range of 80 nm around a central wavelength of 800 nm predicts GVD$_{min} = -264$ fs$^2$ μm$^{-1}$ as a theoretical limit. In practice, the coupling efficiency to the anomalous dispersive mode drops close to cutoff and a perfectly parabolic dispersion relation cannot be achieved, which constrains real devices to approximately half of this limit.

**Compressor coating design and modeling results.** For the final design, we identify promising design parameters (nanopillar diameter, height, and periodicity) by exploring a large parameter space using rigorous coupled-wave analysis (S4)[28]. We then switch to finite-difference time-domain simulations (FDTD, Lumerical FDTD Solutions) and use the parameters (nanopillar radius, height, and unit cell size) as a starting point for fine-tuning the design for large negative GDD, low higher-order dispersion, and high transmission over an extended spectral band. The resulting compressor geometry (Fig. 1b) shows strong anomalous GDD centered at 800 nm wavelength; its phase, group delay, and transmission characteristics are shown in Fig. 2a–c. From 760 to 840 nm, the group delay deviates <2 fs from a group delay profile with constant anomalous GDD of −71 fs$^2$. If a highly linear group delay profile is required, a stronger-dispersive regime exists between 780 and 825 nm: it provides anomalous GDD = −82 fs$^2$ with <0.5 fs deviation from a linear group delay profile. FDTD modeling predicts close to unity

transmission over the full working range. The obtained GDD values are comparable with many commercially available chirped mirrors. By using coating thicknesses of up to 10 μm, highly dispersive chirped mirrors[29] provide larger absolute anomalous GDD than the presented device over a similar bandwidth. However, the presented compressor coating excels when comparing the induced GDD per coating thickness.

The response of the compressor's dispersion and transmission properties to changes of the nanopillar diameter, height, and periodicity is explored in Supplementary Figs. 1 and 2. By changing the nanopillar diameter, the broadband Mie-type transverse magnetic dipole resonance[19] of the nanopillars can be spectrally shifted to achieve anomalous dispersion at a desired central wavelength. The periodicity (475 nm × 475 nm) of the nanopillar array is chosen smaller than the smallest operation wavelength in air and the substrate material ($\lambda^{min}_{SiO_2} = \frac{\lambda^{min}_{Air}}{n_{SiO_2}} = \frac{760\,\text{nm}}{1.45} \approx 524$ nm). Consequently, all diffraction orders except the zeroth order are evanescent outside of the compressor and the incoming beam profile is not modified by the compressor coating[30] (see Supplementary Fig. 3 for the simulated far-field profile). By changing the nanopillar height, the magnitude of the induced anomalous dispersion can be controlled. In practice, only specific nanopillar height ranges can be realized because the coating —similar to a thin-film coating—can be reflective for mismatched height. The design introduced here features an undesirable narrowband Fabry-Perot resonance within the spectral working range (see Fig. 2c). For a given nanopillar diameter and height, the periodicity of the array can be tuned to suppress this resonance (see Supplementary Figs. 1 and 2) and achieve a smooth phase and transmission profile. The interaction between the narrowband Fabry-Perot and the broadband Mie-resonance is described in detail elsewhere[19,20,31,32]. The small residual signature of this resonance is inconsequential for the temporal envelope of broadband ultrashort laser pulses owing to its narrow spectral width. The transmission and phase characteristics of the compressor do not change significantly for slanted incidence angles of up to 5° for s-polarized light (see Supplementary Fig. 4).

**Experimental characterization.** To experimentally verify our design, we fabricated compressors with varying nanopillar

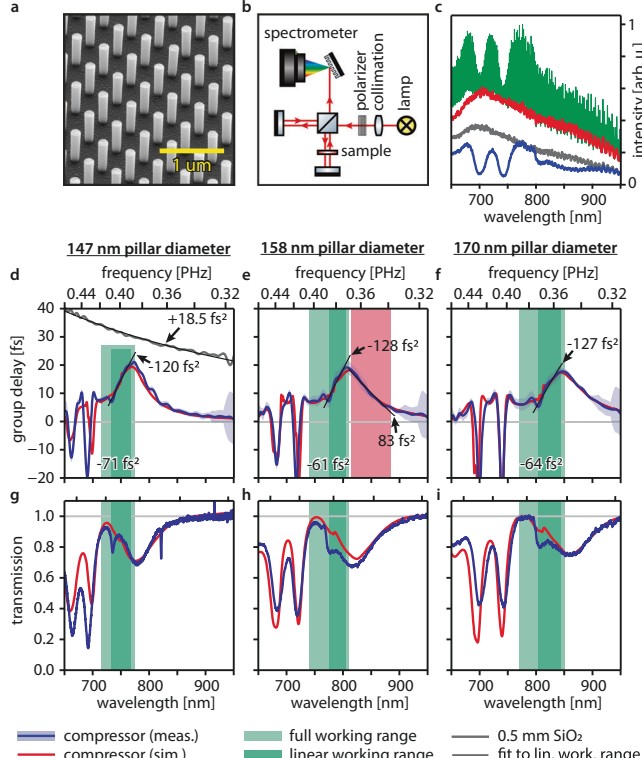

**Table 1 Experimental compressor characterization.**

| Nanopillar diameter [nm] | 147 ± 7 | 158 ± 7 | 170 ± 6 |
|---|---|---|---|
| Full working range [nm] | 705–775 | 740–810 | 770–850 |
| Full working range average GDD [fs²] | −71 ± 1 | −61 ± 2 | −64 ± 2 |
| Linear working range [nm] | 735–770 | 775–805 | 800–840 |
| Linear working range average GDD [fs²] | −120 ± 2 | −128 ± 6 | −127 ± 4 |
| Full working range average transmission | 81% | 79% | 80% |

Figure 3 and Table 1 present the experimental group delay characteristics of compressors for three different nanopillar diameters, measured using white-light interferometry[33] (see methods). The coatings induce anomalous GDD of up to $-71$ fs² over a working bandwidth of up to 80 nm around a center wavelength tunable by changing the nanopillar diameter. This suffices for compensating the GDD of 2 mm-thick fused silica glass (GDD(SiO₂, 800 nm) $= +36.2$ fs² mm⁻¹)[34]. The FDTD predictions match the experimental group delay and transmission profiles well.

We observe a residual signature of the suppressed Fabry-Perot resonance (see above). This significantly limits the linear working range on the high-frequency side (see, e.g., Fig. 3d, g). However, at the same time, it increases the anomalous GDD in the linear working range to up to $-128$ fs², (see black lines in Fig. 3d–f). The GDD magnitude of our compressor makes it ~5800 times more dispersive than glass per unit length. A broadband dip of the transmission profiles on the low-frequency side of the working range decreases the average transmission in the working range to ~80%. This behavior is reproduced when including a 4 nm fabrication tolerance of the nanopillar diameter in the simulations. Thus, close to unity transmission should be achievable by using improved fabrication.

At wavelengths above the working range, the group delay of the realized compressor coatings (see the red area in Fig. 3e) decreases. In this wavelength range, the compressors induce positive GDD of up to 83 fs². Consequently, also compact pulse stretchers can be implemented as frequency-shifted compressor coatings.

**Demonstration of ultrashort laser pulse compression.** To demonstrate the viability of our concept in a real application, we inserted a compressor (nanopillar diameter 162 ± 6 nm) in the path of a mode-locked titanium-sapphire oscillator. To confirm the resulting changes in the femtosecond laser pulses directly in the time domain, we employ second-harmonic frequency-resolved-optical-gating[35] (SH-FROG, see Fig. 4f for setup and methods for details): laser pulses are split into two replicas and delayed with respect to each other. They are then noncollinearly overlapped in a nonlinear crystal. Only when the pulses traverse the crystal simultaneously, second-harmonic radiation is emitted towards the detector. Thus, recording the delay-dependent second-harmonic spectrum yields a spectrogram from which the intensity and phase profiles of the laser pulses can be reconstructed. Because the second-harmonic process mixes two photons from identical pulse copies, the generated spectrum $S(\tau) = S(-\tau)$ is equal for positive and negative delay time $\tau$ between the two pulse copies. Thus, SH-FROG spectrograms are symmetric with respect to the zero-delay time (see, e.g., Fig. 4a, b). Due to this symmetry, the sign of the GDD measured using SH-FROG can be ambiguous[36]. By ensuring that the incoming pulse GDD magnitude exceeds that of the compressor, sign changes of the GDD can be avoided, thus eliminating this uncertainty.

**Fig. 3 White-light interferometry of compressors designed for different operating wavelengths. Samples are labeled by the nanopillar diameter. a** Scanning electron microscope picture of the compressor with 158 nm nanopillar diameter. **b** White-light interferometer setup. Thermal radiation from a tungsten lamp (red lines) is collimated, polarized, and then split in a Michelson interferometer. Light in one interferometer arm passes the sample twice, light in the other arm is delayed. The variable delay between the two arms causes spectral intensity oscillations associated with the compressor group delay dispersion that are recorded by a grating spectrometer. **c** Example of white-light interferometer raw data for the sample with 170 nm nanopillar diameter. Interference spectrum (green); spectrum transmitted by the compressor when the reference arm is blocked (blue) and when the compressor sample is replaced by a fused silica substrate and the reference arm is blocked (gray); the spectrum of the lamp, when the sample arm is blocked (red). **d–f** Compressor group delay profile measured by white-light interferometry (blue lines). FDTD simulations (red lines) account for 4 nm nanopillar diameter fabrication tolerance. The black lines are linear fits to the measured group delay profiles in the linear working ranges (dark green areas) and are labeled with their group delay dispersion (slope). The average group delay dispersions in the full working ranges (light green areas) are also indicated. The red area in **e** marks the spectral region in which the compressor coating imprints positive group delay dispersion. The measured group delay profile of the fused silica substrate (gray line) and a fit to it (black line) are shown for reference. Both were shifted vertically to fit the plotting region. The blue-shaded areas represent the measurement uncertainty: to retrieve an upper bound for the systematic error, we compare a reference measurement on the fused silica substrate with its literature group delay profile. We then use the maximum deviation of the two in a 50 nm bandwidth around each wavelength point as uncertainty and add the standard deviation. **g–i** Measured (blue lines) and simulated (red lines) compressor transmission characteristics relative to the fused silica substrate transmission.

diameters by lithographic top–down processing of a 610 nm-thick amorphous silicon layer on a 0.5 mm-thick fused silica substrate (see methods), Fig. 3a shows a sample imaged using scanning electron microscopy.

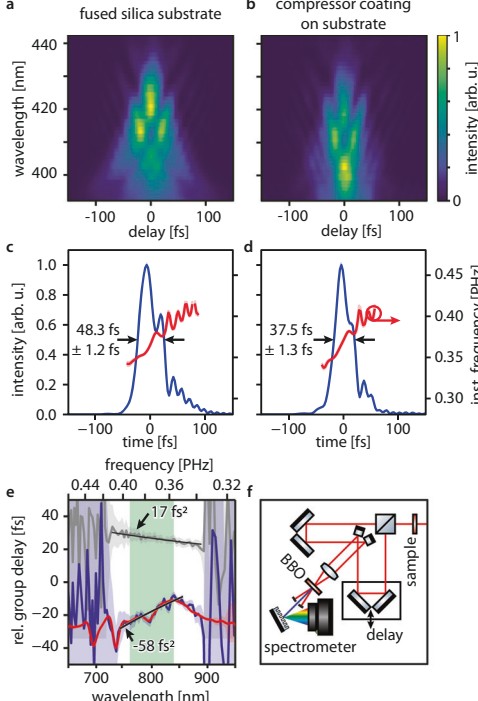

**Fig. 4 Compression of an ultrashort laser pulse. a–b** Experimental second-harmonic frequency-resolved optical-gating (SH-FROG) spectrograms recorded after the incoming laser pulses are transmitted **a** through the fused silica substrate only and **b** through the compressor-coated substrate (nanopillar diameter 162 nm). **c** Time-domain intensity (blue line) and instantaneous frequency (red line) profiles of the laser pulses transmitted through the fused silica substrate retrieved using an iterative ptychographic reconstruction algorithm (see methods). The arrows indicate the pulse full-width at-half maximum duration. Standard deviations (blue and red shaded areas) were determined using the bootstrap method (see methods). **d** Time-domain intensity (blue line) and instantaneous frequency (red line) profiles of the laser pulses transmitted through the compressor-coated substrate. The measured full-width at-half maximum durations show the pulse shortening by 11 fs. **e** group delay profile retrieved from the SH-FROG measurements (SH-FROG measurement: blue line, white-light interferometer measurement of the same sample: red line) and least-squares fit (black line) to the data in the full working range (light green area). The group delay profile retrieved for the fused silica substrate (SH-FROG measurement: gray line) is displayed as a reference. **f** SH-FROG setup. The incoming laser pulses (red lines) are modified by the sample, split and delayed by the arms of an interferometer, and subsequently focused and overlapped noncollinearly in a beta-Barium-Borate (BBO) crystal. Second-harmonic radiation (blue line) generated by combining one photon from each arm is detected in a grating spectrometer for different delay times and reveals the temporal structure of the laser pulses.

We measured spectrograms for: (i) the input laser pulses, (ii) the laser pulses affected by the fused silica substrate, and (iii) the laser pulses after traversing the compressor and substrate. Experimental spectrograms are displayed in Fig. 4a, b, whereas the full data set, retrieved spectrograms, and retrieval process[37,38] are detailed in the methods and Supplementary Fig. 5. Both, visual comparison of the reconstructed spectrograms (see Supplementary Fig. 5) and the trace-area normalized FROG error G' < 0.05[39,40] of the reconstructed spectrograms indicate good agreement.

In the time-domain (Fig. 4c, d), the compressor reduces the pulses' full-width-at-half-maximum duration by a quarter from

(48.3 ± 1.2) fs to (37.5 ± 1.3) fs, clearly demonstrating compression although the laser bandwidth exceeds the linear compressor working range. We can relate these time-domain measurements with the group delay profiles obtained from white-light interferometry up to a constant offset (Fig. 4e). Agreement within the uncertainty proves both methods are viable for the determination of the group delay properties of thin compressor coatings. Spectral-domain results are summarized in Tables 2 and 3, Fig. 4e, and Supplementary Fig. 5. When numerically applying the measured characteristics to a light pulse matched to the full working range (Gaussian-shape, FWHM spectral bandwidth 760 nm–840 nm) stretched from 10 fs to 19 fs by 1.6 mm-thick fused silica glass, the compressor suppresses >65% of the pulse elongation and recompresses the pulse to below 13 fs. Thus, it can sufficiently compensate for the dispersion of many beam splitters, polarizers, waveplates, et cetera.

We demonstrated transmissive broadband pulse-compressor nanocoatings for the important visible to near-infrared spectral region, which compensate the GDD of up to 2 mm-thick fused silica glass over a bandwidth of up to 80 nm, and their application to femtosecond light pulses. Our approach can be implemented on conventional optics and requires no spatial, angular, or polarization pre-conditioning of the incoming light, therefore it can be rapidly implemented in optical setups. As group delay characteristics are determined by geometric properties, rather than material dispersion, the approach is flexible and can be adapted to different spectral regions or applications. In the future, stacked or more intricate structures tailored by inverse design or machine learning can expand the technique towards engineering complex pulse shapes for coherently controlling chemical reactions and quantum systems or optimizing nonlinear processes such as high-harmonic generation.

## Methods

**Group-velocity dispersion limit**. To determine a lower limit for the constant anomalous GVD achievable for a given working bandwidth in a slow-light compressor coating, we start by parametrizing the dispersion relation in the working region as a parabola $k_z(\omega) = \frac{1}{2}\text{GVD}(\omega - d)^2 + e$ with parameters GVD, $d$ and $e$. This enforces a constant $\text{GVD} = \frac{d^2 k_z}{d\omega^2}$ by definition. Using that the maximum group velocity in the compressor coating should be realized at the high-frequency side of the working range, but cannot exceed the speed of light $\frac{1}{v_g}\big|_{\omega=\omega^+} = \frac{dk_z}{d\omega}\big|_{\omega=\omega^+} = \frac{1}{c}$, fixes $d$ and yields

$$k_z(\omega) = \frac{1}{2}\text{GVD}\left(\omega + \frac{1}{\text{GVD}*c} - \omega^+\right)^2 + e \qquad (2)$$

Furthermore, fixing the wavevector at the limits of the working range, $k_z(\omega^+) = \Delta k$ and $k_z(\omega^-) = 0$ and eliminating $e$ from Eq. (2) connects the lower limit for the constant anomalous group-velocity dispersion with the working bandwidth and the wavevector change $\Delta k$ over that bandwidth:

$$\text{GVD}_{\min} = \frac{2}{\Delta\omega}\left(\frac{1}{c} - \frac{\Delta k}{\Delta\omega}\right) \qquad (3)$$

Using the definition of the (effective) refractive index $n = \frac{k_z * c}{\omega}$, and $n^+ = n(\omega^+) = \frac{k_z * c}{\omega}\big|_{\omega=\omega^+}$ yields $\Delta k = \frac{n^+ * \omega^+}{c}$ and combining with Eq. (3) yields Eq. (1).

**Fabrication**. First, we deposit a 610 nm-thick amorphous silicon layer on a 500 μm-thick fused silica substrate using plasma-enhanced chemical vapor deposition. After spin-coating a layer of negative electron beam resist (Micro Resist Technology, ma-N 2403) and an additional layer of conductive polymer (Showa Denko, ESPACER 300) to avoid charging effects during electron beam lithography (EBL), we define the nanopillar mask patterns using EBL (Elionix, ELS-F125) and develop (MicroChemicals, MIF 726). Anisotropic inductively coupled plasma-reactive ion etching (ICP-RIE using a mixture of SF6 and C4F8) was used to etch the nanopillar structures. The electron beam resist mask was removed by immersing the sample in piranha solution.

The demonstrated compressor coatings cover an area of 3.1 mm². Both, the employed material system and feature sizes are manufacturable via commercially available deep-ultraviolet (immersion) lithography tools. Thus, high quantities and large-scale devices can straightforwardly be manufactured using prevalent industrial semiconductor manufacturing methods.

**Table 2 Femtosecond pulse compression: laser pulse GDD in the full working range (760–840 nm).**

| Incoming pulse | Substrate | Compressor and substrate |
|---|---|---|
| $(+191 \pm 5)$ fs$^2$ | $(+210 \pm 4)$ fs$^2$ | $(+154 \pm 4)$ fs$^2$ |

**Table 3 Femtosecond pulse compression: optics GDD in the full working range (760–840 nm).**

| Substrate | Compressor |
|---|---|
| $(+17 \pm 6)$ fs$^2$ | $(-58 \pm 6)$ fs$^2$ |

**White-light interferometry**. We determine the group delay profiles of our compressors using a self-built white-light interferometer. Broadband black body radiation from a tungsten lamp (Thorlabs SLS202L) is coupled to multimode fiber, then collimated and linearly polarized before a Michelson interferometer. In the interferometer, light in one arm is used as a reference and is delayed, light in the other arm is modified by the sample. After the interferometer, the spectral interference between light from both arms is resolved using an Andor Shamrock spectrometer and reveals the group delay profile of the sample.

**Femtosecond oscillator, FROG measurements, and retrieval**. Measurements were performed on the uncompressed output of a Femtosource Rainbow (FEM-TOLASERS Produktions GmbH) femtosecond oscillator using a self-built non-collinear SH-FROG. The uncompressed pulses are elongated by the GDD and third-order dispersion of the oscillator, a fused silica lens, beam splitter, and air path (spectral phase of the incoming pulses see Supplementary Fig. 5j). In the time domain (Supplementary Fig. 5g), the GDD causes a symmetric broadening, whereas the third-order dispersion causes the substructure. The group delay profile shown in Fig. 4e is determined by subtracting the group delay profiles in Supplementary Fig. 5k, l. Because these were both measured using the same incoming laser pulses, the subtraction eliminates the group delay profiles of the incoming laser pulses. We employ a self-written retrieval algorithm based on the iterative ptychographic engine to retrieve the pulse characteristics (Supplementary Fig. 5g–l) from the experimental spectrograms (Supplementary Fig. 5a–c). We correct for the phase-matching conditions in our 100 µm-thick beta-barium-borate crystal before the retrieval and ignore non-phase-matched spectral components during the retrieval (see Supplementary Fig. 5a–c). To increase the accuracy of the retrieval, we use the measured power spectrum of the fundamental laser pulses as an additional constraint. This approach retrieves correct temporal and spectral pulse properties, even for incomplete spectrograms. In our case, it provides reliable spectral information even in the non-phase-matched spectral ranges of the ultrashort pulses, witnessed by the excellent agreement between the compressor's group delay profile measured using the SH-FROG and white-light interferometry (Fig. 4e). Spectrograms calculated from the retrieved pulse parameters are displayed in Supplementary Fig. 5d–f for comparison. Errors were determined using the bootstrap method.

## Data availability

The data supporting the findings of this study are available in figshare with the identifier https://doi.org/10.6084/m9.figshare.16589867 or from the corresponding authors upon request.

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

## Acknowledgements

This work was performed, in part, at the Center for Nanoscale Systems (CNS), a member of the National Nanotechnology Coordinated Infrastructure (NNCI), which is supported by the NSF under award no. ECCS-2025158. CNS is a part of Harvard University. M.O. acknowledges a Feodor-Lynen Fellowship from the Alexander von Humboldt Foundation. Y.-W.H. acknowledges support from the Ministry of Science and Technology, Taiwan (grant no. 110-2124-M-A49-004), support from the Ministry of Education (MOE) in Taiwan under the Yushan Young Scholar Program, and the Higher Education Sprout Project of the National Yang Ming Chiao Tung University. Z.W. acknowledges funding by the China Scholarship Council (201906180074). Additionally, financial support from the Office of Naval Research (ONR), under the MURI program, grant no. N00014-20-1-2450, and from the Air Force Office of Scientific Research (AFOSR), under grant no. FA95550-19-1-0135, is acknowledged.

## Author contributions

M.O., W.T. C., Y.A.I. and X.Y. carried out the numerical simulations and designed the final compressor coatings. Y.-W.H. fabricated the compressor coatings. M.O. and Z.W. carried out the experimental characterization and ultrashort pulse compression. M.O. and X.Y. analyzed the experimental and simulated data. M.O., M.S. and F.C. wrote the manuscript. F.C. supervised the study. All authors discussed the manuscript.

## Competing interests

The authors declare no competing interests.
