## [Peer Review File · Nature Communications]

Slow Light Nanocoatings for Ultrashort Pulse CompressionREVIEWER COMMENTS

Reviewer #1 (Remarks to the Author):

In the submitted work, the authors proposed a concept of high-aspect-ratio nanopillar metasurface-based ultrafast pulse compressors and performed proof-of-concept measurements on carefully designed and fabricated devices. The operating principle is based on coupling of the incident ultrashort pulse to a slow light mode of the nanopillar array (as in a coupled waveguide array). Although the proposed idea is very straightforward and the extendibility seems to be rather limited (i.e., at the current stage, to a simple pulse compressor), I think that the work is interesting as the authors provide a basic set of guidelines as a starting point for designing a metasurface-based pulse shaping device. Therefore, with all aforementioned considerations, I recommend the publication of this work in Nature Communications. However, the following suggestions need to be addressed to make the manuscript be in better shape.

(1) I suggest the authors change the title of the manuscript; as the authors demonstrated only the pulse compression (and not the stretching or more sophisticated shaping in general), a more accurate title would be “Slow Light Nanocoatings for Ultrashort Pulse Compression”. From the data shown in the manuscript, a simple stretching functionality seems to be quite easily achievable. However, it is also certain that those are not explicitly discussed or demonstrated experimentally.

(2) In my opinion, the working principle section would be better to be relocated in earlier sections of the manuscript (with some modifications in the figure panels).

(3) In Fig. 4(b), the power fractions at higher frequencies do not add up to unity. This seems to be attributed to the existence of higher order modes. It would be a little bit better to show a certain number of contributing higher order modes that are matched to the input excitation by symmetry. In addition, please show reflection and absorption in Fig. 1 or 2, which would be informative and self-contained for better understanding.

Reviewer #2 (Remarks to the Author):

Please, find attached my review as a pdf file.

Reviewer #3 (Remarks to the Author):

In this work Ossiander et al proposed a nano-patterning approach for achieving dispersion compensation and consequently Femto-second laser pulse compression. The nanocoating was made from array of amorphous silicon nano-pillars. By carefully engineering the height, diameter of the nanopillars and the periodicity of the nanopillar array, the authors were able to control the anomalous group delay dispersion of the nanocoating at desired wavelength range. They experimentally demonstrate the compression of a laser pulse from 48 fs to 37 fs relative to an uncoated glass wafer. The underlying mechanism is the presence of a HE₁₂-like mode with strong anomalous group delay features close to its cut-off frequency. The work may lead to some useful applications of nano-patterned coating in ultrafast optics. However, before I can make a recommendation the authors need to address the following questions.

1. The authors should benchmark the performance of their approach with the well-established traditional approaches.
2. For even shorter pulses, the operation bandwidth needs be further broadened. What is the upper limit of the operation bandwidth at a given central wavelength?
3. On page 3, line 74: regarding “only specific nanopillar heights can be realized because Fabry-Perot-resonances have to be accounted for to achieve high transmission” , does the Fabry Perot resonance mentioned here correspond to the sharp spectral feature in Fig. 1d?

Specific comments and answers to the reviewers

In the following, we give a one-to-one response (printed in blue) to the individual reviewer's comments which are printed in green.

Reviewer #1 (Remarks to the Author):

(1) I suggest the authors change the title of the manuscript; as the authors demonstrated only the pulse compression (and not the stretching or more sophisticated shaping in general), a more accurate title would be "Slow Light Nanocoatings for Ultrashort Pulse Compression". From the data shown in the manuscript, a simple stretching functionality seems to be quite easily achievable. However, it is also certain that those are not explicitly discussed or demonstrated experimentally.

We changed the title as the reviewer suggested. As she/he states correctly, pulse stretching is easy to obtain using this nanodevice class by shifting the maximum of the induced group delay and using the spectral range on the long-wavelength side of the maximum. The experimental group delay profiles determined via white-light interferometry already show this stretching group delay behavior in the realized samples. We added a paragraph to the manuscript and the red shaded region to Fig. 3e (previously Fig. 2e) to highlight this possibility:

At wavelengths above the working range, the group delay of the realized compressor coatings (see red area in Fig. 3e) decreases. In this wavelength range, the compressors induce positive GDD of up to 83 fs^2 . Consequently, also compact pulse stretchers can be implemented as frequency shifted compressor coatings.

(2) In my opinion, the working principle section would be better to be relocated in earlier sections of the manuscript (with some modifications in the figure panels).

We moved the working principle section to the beginning and adapted the figures.

(3a) In Fig. 4(b), the power fractions at higher frequencies do not add up to unity. This seems to be attributed to the existence of higher order modes. It would be a little bit better to show a certain number of contributing higher order modes that are matched to the input excitation by symmetry.

The reviewer is right, in the initial version of the manuscript the power fractions did not add up to unity. This behavior was however not a signature of higher modes in the compressor: at high frequencies, silicon starts absorbing and the displayed power fractions were not corrected for this absorption. At low frequencies (directly below the working range), the coupled power fractions do not add up to unity because of a reflection at the front facet of the coating.

To clarify the manuscript, the new Fig. 1e (previously Fig. 4b) is corrected for absorption at high frequencies and displays the reflection from the front facet. The power fractions add up to unity by only including the two mentioned modes and the reflection.

(3b) In addition, please show reflection and absorption in Fig. 1 or 2, which would be informative and self-contained for better understanding.

We agree this is valuable information and added the reflection and absorption of the compressor coating to the new Fig. 2c (previously Fig. 1e).

Reviewer #2 (Remarks to the Author):

1. I would like to understand better how is the proposed practical implementation. Let's imagine I do have an ultrashort pulse laser and I do have the issues with pulse chirping with a lens. Now, how much should I coat my lens with the periodic array? The entire width of the pulse, I imagine. That would be like an area of about mm^2 . Isn't it rather challenging? I don't think this issue is discussed in the manuscript.

The reviewer is right, the device must be coated with a nanopillar array that exceeds the desired beam spot size. With current nanofabrication technology, this does not pose significant constraints: the experimentally realized devices in our manuscript already cover an area of 3.1 mm^2 and were manufactured using conventional prototyping electron beam lithography tools. Even using older prototyping systems, such devices can be manufactured within less than 2 hours of write time. However, for commercialization, one would use much higher throughput systems such as (immersion) deep-UV lithography, which has been shown to produce high quality optical devices with feature sizes comparable to those used in the compressor coating (see, e.g., Hu et al. Optics Express 26, 15, pp. 19548-19554 (2018), <https://doi.org/10.1364/OE.26.019548>) over as large as 12-inch wafers in exposure times of less than a minute.

We added this information and reference to the fabrication section:

The demonstrated compressor coatings cover an area of 3.1 mm^2 . Both, the employed material system and feature sizes are manufacturable via commercially available deep-ultraviolet (immersion) lithography^{39, 40} tools. Thus, high quantities and large-scale devices can straightforwardly be manufactured using prevalent industrial semiconductor manufacturing methods.

2. What is the role of disorder in the lattice? If you have to deposit an array of mm^2 with a periodic lattice of about 500 nm, I'm sure there will be imperfections and distortions in the lattice. How is that going to affect the compensation mechanisms? The eigenmodes are going to shift their frequency and, given their strong dispersion relation, I would expect a significant or non-negligible effect.

The spectral behavior of the compressor is stable against small changes of the lattice period, see, e.g., Supplementary Fig. 1f which displays the phase behavior of the compressor coating with changing lattice period. Furthermore, as mentioned above, the experimental devices demonstrated in the paper were up to 3.1 mm^2 in size and were written using 60 individual electron beam lithography write fields. Therefore, any detrimental effects arising from writing large devices, i.e. stitching- or write-field-distortion errors common in electron beam lithography would already be included in the presented experimental data. Because the lattice period (i.e., long range order) is only dependent on the precision of the exposure tool, it is generally much more robust than the local feature size, which we identified in the manuscript as the source of the small deviation between the modeled and experimentally observed performance (initially Fig. 2, now Fig. 3).

3. How much is the absorption of the coat? We are talking about 1 micron's height silicon pillars working in the near-infrared. Short pulses concentrate a rather high power in time so I would also expect an effect of silicon absorption.

We added the calculated absorption of the compressor coating to the new Fig. 2c (previously Fig. 1e) of the revised manuscript (see also answer below). Because less than 30% of the power is transmitted in the silicon pillars and the absorption of silicon is low in the near infrared (see, e.g., Schinke et al., AIP Advances 5, 067168 (2015); <https://doi.org/10.1063/1.4923379>), our simulations predict < 1% linear absorption in the working range. Nonlinear absorption is not a significant source of attenuation because optical components in ultrafast lasers are rarely located close to foci due to damage threshold

constraints. During normal propagation, even for a chirped-pulse-amplified femtosecond laser (e.g., 1 mJ pulse energy, 10 fs pulse duration, and a Gaussian beam profile with 15 mm FWHM diameter), the maximum pulse intensity I remains below 50 GW cm^{-2} . We modeled the nonlinear absorption for this case using the two-photon absorption coefficient of silicon, $\beta \approx 2 \text{ cm GW}^{-1}$ (Bristow et al., Appl. Phys. Lett. 90, 191104 (2007); <https://doi.org/10.1063/1.2737359>) and including field enhancement in the compressor and find $< 1\%$ nonlinear absorption.

We added this information to the revised manuscript:

Two-photon absorption⁴⁴ in the compressor coating remains below 1% for incident intensities beyond 50 GW cm^{-2} .

I also would like to understand Fig. 1e. The transmission plotted in the figure is absolute? Relative to anything? Because it seems to get above 1 for some frequencies (2.0 PHz or 2.5 PHz). How much light do I lose if I coat my lens with this coating? That's something important to discuss.

The reviewer's observation was right, the transmission plotted in Fig. 1e (now Fig. 2c) exceeded unity slightly. This was because the transmission plotted in Fig. 1e (now Fig. 2c) used the same normalization as the transmissions plotted in Fig. 2 (now Fig. 3), i.e., it was measured relative to the uncoated SiO_2 substrate. Because the compressor acts as an antireflection coating, this led to above-unity transmission. Unfortunately, only the caption of Fig. 2 (now Fig. 3) mentioned this normalization. Coating a lens with an ideal coating would thus not lose light compared to an uncoated lens. Fig. 3 (previously Fig. 2) displays the experimentally achieved transmissions.

To clarify the revised version, we changed Fig. 1e (now Fig. 2c) to show the absolute transmission and in view of reviewer #1 also the reflection and absorption characteristics of the compressor coating.

4. Another issue that is not fully clear to me is the far-field profile of the mode out coupled from the coat. Is there any diffraction effect due to the presence of the coat? Is there any deviation of the beam induced by the coat? I find this particularly relevant when using carefully aligned beams in an optical setup where any tiny spatial deviation or diffraction may have a potentially disrupting effect.

As the periodicity ($475 \text{ nm} \times 475 \text{ nm}$) of the compressor coating is substantially lower than all wavelengths in the working range and uniform, the far field pattern of the compressed laser beam remains unchanged. Applying the coating to an optics causes neither diffraction orders nor beam deviation (except - if crudely angle-misaligned - the displacement any parallel plate with refractive index $\neq 1$ would cause).

To emphasize this point, we added the beam profile of a Gaussian beam transmitted through the compressor coating as Supplementary Fig. 3 and a paragraph to the revised manuscript:

The periodicity ($475 \text{ nm} \times 475 \text{ nm}$) of the nanopillar array is chosen smaller than the smallest operation wavelength in air and the substrate material ($\lambda_{\text{SiO}_2}^{\text{min}} = \frac{\lambda_{\text{Air}}^{\text{min}}}{n_{\text{SiO}_2}} = \frac{760 \text{ nm}}{1.45} \approx 524 \text{ nm}$). Consequently, all diffraction orders except the zeroth order are evanescent outside of the compressor and the incoming beam profile is not modified by the compressor coating²⁸ (see Supplementary Fig. 3 for the simulated far field profile).

5. There is an issue with the Fabry – Perot cavities due to the periodicity which is mitigated by tuning different parameters in the system. However, there is a residual signal. How does this residual affect the pulse compression? Is completely negligible?

To check the influence of the residual feature, we simulated the compression of a Gaussian pulse matched to the working range and elongated by the propagation through fused silica using both: the FDTD-modeled group delay profile and the experimentally measured group delay profile. If we correct the residual feature in these profiles and compress using the corrected profiles instead, the final pulse duration changes by less than 0.05 fs in both cases, which we consider negligible for real-world applications.

6. As final small remark, I find the use of angular frequencies a bit unclear. Maybe the authors would use natural frequencies instead?

We changed all figures to use natural frequency instead of angular frequency.

Reviewer #3 (Remarks to the Author):

1. The authors should benchmark the performance of their approach with the well-established traditional approaches

Established techniques providing anomalous dispersion for the temporal compression of ultrafast laser sources are prism/grating compressors and chirped mirrors. As we point out in the manuscript, we believe that the working principle of our slow light approach is distinctly different from these techniques because it is working in transmission and without altering the light propagation direction. We are unaware of traditional approaches achieving negative dispersion in the visible frequency range in transmission without additional beam path and optics. As we demonstrate experimentally, our coating can compensate for the dispersion a typical optical element (beam splitter, lens, ...) introduces. In an ideal application, such a coating would act in an anti-dispersive manner in analogy to anti-reflective coatings that are standard in laser science. Such coatings could waive the need for additional prism/grating compressor setups in an ultrafast laser and provide significant extra flexibility in the design and test of novel setups. As each element would be equipped with its individual dispersion compensation, achieving a maximal amount of GDD is secondary.

To highlight this aspect better we amended the introductory paragraph:

A key challenge in their application is dispersion control: Because all transparent materials are normally dispersive in the ultraviolet, visible, and near-infrared regions below 1.3 μm , the realization of compressed laser pulses currently requires complex angular dispersive⁵⁻⁷, reflective⁸⁻¹⁰, or photonic-crystal-fiber-based compression setups¹¹, which all add significant complexity, path length, and beam deviations to the optical setup.

To benchmark the performance of the slow-light approach demonstrated here we included the comparison to commercially available chirped mirror technology:

The obtained GDD values are comparable with many commercially available chirped mirrors. By using coating thicknesses of up to 10 μm , highly dispersive chirped mirrors²⁸ provide larger absolute anomalous GDD than the presented device over a similar bandwidth. However, the presented compressor coating excels when comparing the induced GDD per coating thickness.

2. For even shorter pulses, the operation bandwidth needs be further broadened. What is the upper limit of the operation bandwidth at a given central wavelength?

We agree this is an interesting limit and good addition to the current manuscript. We included a new discussion at the end of the working principle section in the manuscript:

A lower limit GDD_{\min} for the achievable constant anomalous GDD in a transmissive slow-light coating can be estimated similar to the maximum achievable delay in slow light waveguides or scatterers^{26, 27}. For a given working range $\omega \in \left] \omega_0 - \frac{\Delta\omega}{2}, \omega_0 + \frac{\Delta\omega}{2} \right] =]\omega^-, \omega^+]$ defined by the cutoff $k_z(\omega^-) = 0$ and bandwidth $\Delta\omega$, we find (see methods) that the achievable constant anomalous GDD is set by the thickness of the coating L and the effective refractive index of the HE12 mode on the high-frequency side of the working range $n^+ = n(\omega^+) = \frac{k_z * c}{\omega} \Big|_{\omega=\omega^+}$ (vacuum speed of light c):

$$GDD_{\min} = L * GVD_{\min} = \frac{L}{c\Delta\omega} \left(2 - n^+ - 2 \frac{n^+ * \omega_0}{\Delta\omega} \right) \quad (1)$$

As example, using $n^+ \approx 1$ as suggested by Fig. 1c and choosing a working range of 80 nm around a central wavelength of 800 nm predicts $GVD_{\min} = -264 \text{ fs}^2 \text{ um}^{-1}$ as theoretical limit. In practice, the coupling efficiency to the anomalous dispersive mode drops close to cutoff and a perfectly parabolic dispersion relation cannot be achieved, which constrains real devices to approximately half of this limit.

and in the methods:

To determine a lower limit for the constant anomalous GVD achievable for a given working bandwidth in a slow-light compressor coating, we start by parametrizing the dispersion relation in the working region as a parabola $k_z(\omega) = \frac{1}{2}GVD(\omega - d)^2 + e$ with parameters GVD, d and e . This enforces a constant $GVD = \frac{d^2 k_z}{d\omega^2}$ by definition. Using that the maximum group velocity in the compressor coating should be realized at the high frequency side of the working range, but cannot exceed the speed of light $\frac{1}{v_g} \Big|_{\omega=\omega^+} = \frac{dk_z}{d\omega} \Big|_{\omega=\omega^+} = \frac{1}{c}$, fixes d and yields

$$k_z(\omega) = \frac{1}{2}GVD \left(\omega + \frac{1}{GVD * c} - \omega^+ \right)^2 + e. \quad (2)$$

Furthermore, fixing the wavevector at the limits of the working range, $k_z(\omega^+) = \Delta k$ and $k_z(\omega^-) = 0$ and eliminating e from eq. (2) connects the lower limit for the constant anomalous group velocity dispersion with the working bandwidth and the wavevector change Δk over that bandwidth:

$$GVD_{\min} = \frac{2}{\Delta\omega} \left(\frac{1}{c} - \frac{\Delta k}{\Delta\omega} \right). \quad (3)$$

Using the definition of the (effective) refractive index $n = \frac{k_z * c}{\omega}$, and $n^+ = n(\omega^+) = \frac{k_z * c}{\omega} \Big|_{\omega=\omega^+}$ yields $\Delta k = \frac{n^+ * \omega^+}{c}$ and combining with eq. (3) yields eq. (1).

3. On page 3, line 74: regarding “only specific nanopillar heights can be realized because Fabry-Perot-resonances have to be accounted for to achieve high transmission” , does the Fabry Perot resonance mentioned here correspond to the sharp spectral feature in Fig. 1d?

The reviewer is right, one possible signature of a mismatched height is the is the sharp feature marked with the purple arrow in Fig 1d. However, the sentence was meant to convey the broader message that the height influences the overall achievable transmission, much like the thickness of an anti-reflection coating would. We see that the old paragraph structure was not optimal and expanded:

By changing the nanopillar height, the magnitude of the induced anomalous dispersion can be controlled. In practice, only specific nanopillar height ranges can be realized because the coating – similar to a thin-film coating - can be reflective for mismatched height.

REVIEWERS' COMMENTS

Reviewer #1 (Remarks to the Author):

Dear Editor,

I read the authors' response to my comments. I have no further questions and agree on the publication of the manuscript.

Regards,

Bumki Min

Reviewer #2 (Remarks to the Author):

I have read carefully the reply to my comments and I am satisfied with the additional information given by the authors. I recommend publication of the updated manuscript.

Reviewer #3 (Remarks to the Author):

The authors have addressed all the points raised in my previous report. I am happy to recommend it for publication in Nature Communications.